# Rate of reclassification of HER2-equivocal breast cancer cases to HER2-negative per the 2018 ASCO/CAP guidelines and response of HER2-equivocal cases to anti-HER2 therapy

James Crespo[1¤a], Hongxia Sun[2¤b], Jimin Wu[3], Qing-Qing Ding[2], Guilin Tang[4], Melissa K. Robinson[2], Hui Chen[2], Aysegul A. Sahin[2], Bora Lim[1]*

1 Department of Breast Medical Oncology, The University of Texas MD Anderson Cancer Center, Houston, Texas, United States of America, 2 Department of Pathology, The University of Texas MD Anderson Cancer Center, Houston, Texas, United States of America, 3 Department of Biostatistics, The University of Texas MD Anderson Cancer Center, Houston, Texas, United States of America, 4 Department of Hematopathology, The University of Texas MD Anderson Cancer Center, Houston, Texas, United States of America

¤a Current address: Department of Medical Oncology, Americas Centro de Oncologia Integrado, Rio de Janeiro, Brazil
¤b Current address: Department of Breast and Cytopathology, The University of Texas Health Science Center at Houston, Houston, Texas, United States of America
* blim@mdanderson.org

## Abstract

### Purpose

The 2018 American Society of Clinical Oncology/College of American Pathologists (ASCO/CAP) guideline on HER2 testing in breast cancer permits reclassification of cases with HER2-equivocal results by FISH. The impact of such reclassification is unclear. We sought to determine the proportion of HER2-equivocal cases that are reclassified as HER2-negative and the impact of anti-HER2 therapy on survival in HER2-equivocal cases.

### Methods

We reviewed medical records of breast cancer patients who had HER2 testing by fluorescence in stitu hybridization (FISH) and immunohistochemistry (IHC) performed or verified at The University of Texas MD Anderson Cancer Center during April 2014 through March 2018 and had equivocal results according to the 2013 ASCO/CAP guideline. The population was divided into 2 cohorts according to whether the biopsy specimen analyzed came from primary or from recurrent or metastatic disease. HER2 status was reclassified according to the 2018 ASCO/CAP guideline. Overall survival (OS) and event-free survival (EFS) were calculated using the Kaplan-Meier method, and the relationship between anti-HER2 therapy and clinical outcomes was assessed.

### Results

We identified 139 cases with HER2-equivocal results according to the 2013 ASCO/CAP guideline: 90 cases of primary disease and 49 cases of recurrent/metastatic disease. Per

**Data Availability Statement:** All relevant data are within the paper and its Supporting Information files.

**Funding:** The authors received no specific funding for this work.

**Competing interests:** The authors have declared that no competing interests exist.

the 2018 ASCO/CAP guideline, these cases were classified as follows: overall, HER2-negative 112 cases (80%), HER2-positive 1 (1%), and unknown 26 (19%); primary cohort, HER2-negative 85 (94%), HER2-positive 1 (1%), unknown 4 (4%); and recurrent/metastatic, HER2-negative 27 (55%) and unknown 22 (45%). Five patients in the primary-disease cohort and 1 patient in the recurrent/metastatic-disease cohort received anti-HER2 therapy. There was no significant association between anti-HER2 therapy and OS or EFS in either cohort (primary disease: OS, p = 0.67; EFS, p = 0.49; recurrent/metastatic-disease, OS, p = 0.61; EFS, p = 0.78.

## Conclusions

The majority of HER2-equivocal breast cancer cases were reclassified as HER2-negative per the 2018 ASCO/CAP guideline. No association between anti-HER2 therapy and OS or EFS was observed. HER2-equivocal cases seem to have clinical behavior similar to that of HER2-negative breast cancers.

## Introduction

Breast cancer is the most common cancer diagnosed among women worldwide and remains the principal cause of cancer death among women worldwide [1]. Amplification and/or over-expression of the *ERBB2* gene, which encodes human epidermal growth factor receptor 2 (HER2), is seen in 15–20% of invasive breast cancers [2] as determined using fluorescence in situ hybridization (FISH) or immunohistochemical (IHC) staining of the HER2 [2, 3]. HER2 contributes to the aggressive features of breast cancers; however, rapid evolution of anti-HER2 therapies, including anti-HER2 antibodies such as trastuzumab [4], pertuzumab [5], T-DM1 [6], and DS-8201 [7] and pan-HER2 inhibitors such as neratinib [8], lapatinib [9], and tucatinib [10], has significantly improved the survival of patients with HER2-positive disease. Thus, accurate HER2 test results are imperative to identify candidates for these therapies.

Implementation of the 2013 American Society of Clinical Oncology/College of American Pathologists (ASCO/CAP) guideline on HER2 testing in breast cancer, compared with the previous guideline from 2007, resulted in increases in the proportion of HER2-positive breast cancers and the proportion of cases with HER2-equivocal results [11–14]. The increase in cases with HER2-equivocal results is a clinical challenge because the benefit of anti-HER2 therapy for this subgroup is uncertain [2]. The 2018 ASCO/CAP guideline attempted to decrease the number of cases with equivocal results by creating 5 groups based on both FISH and IHC results [15]. The "HER2 equivocal" group in the 2013 ASCO/CAP guideline corresponds to group 4 in the 2018 guideline and is defined by a FISH HER2/CEP17 ratio of <2 and a mean HER2 copy number of ≥4 and <6 signals/cell. The 2018 ASCO/CAP guideline permits cases in group 4 to be further classified as HER2-positive or HER2-negative based on IHC staining for HER2 done using sections from the same tissue sample used for FISH. If the IHC score is 0 or 1+, the diagnosis is HER2-negative. If the IHC score is 3+, the diagnosis is HER2-positive. If the IHC score is 2+, an FISH recount by an additional pathologist who counts at least 20 cells included with the area with IHC 2+ staining is required. If FISH shows a HER2/CEP17 ratio of ≥4, the diagnosis is HER2-positive; otherwise the diagnosis is HER2-negative.

The therapeutic impact of this change between the 2013 and 2018 ASCO/CAP guidelines has yet to be evaluated. In this study, we sought to determine the reclassification rate of HER2-equivocal cases using the new 2018 guideline and the impact of such reclassification in

terms of therapy efficacy and long-term clinical outcomes, using a single-center patient case analysis.

## Materials and methods

### Patients

We reviewed the breast medical oncology (BMO) database to identify breast cancer cases with HER2 FISH performed at The University of Texas MD Anderson Cancer Center between April 2014 and March 2018 and equivocal HER2 results according to the 2013 ASCO/CAP guideline. Not all the cases were treated at MD Anderson, but all cases were reviewed by our own pathologies, and available treatment and follow-up data were captured. If more than 1 HER2-equivocal FISH result occurred in the same patient, we included only the equivocal FISH result with the earliest biopsy date. We excluded cases with a HER2-positive FISH result from testing performed at an outside facility, a HER2-equivocal FISH result from an anatomic site other than the breast, synchronous malignances (except basal cell carcinoma), missing survival data, or receipt of the vaccine E75 because this vaccine was intended to boost the efficacy of anti-HER2 therapy in HER2-equivocal cases. For the remaining cases, we collected clinico-pathologic data, treatments received, and clinical outcomes from medical records in the MD Anderson BMO database.

The selected cases were divided into 2 cohorts according to the origin of the tissue that showed the HER2-equivocal FISH result. The first cohort, designated the primary cohort, included cases with tissue obtained from the breast and/or axilla. These cases had been treated with curative intent with neoadjuvant and/or adjuvant systemic treatment, surgery, and radiotherapy. The second cohort, designated the recurrent/metastatic cohort, included cases with tissue obtained from a site of recurrent or metastatic disease. Almost all the cases in this cohort had been treated palliatively; only a few cases had been treated with curative intent.

All patients with samples included in this study had been enrolled to a Breast Medical Oncology (BMO) registry protocol approved by Institutional Review Board (protocol #2004–0541). BMO database is updated on every 3 months basis to track the recurrence and deaths/other major events of patients who are seen at Nellie Connally Breast Cancer Clinic at MD Anderson cancer center, and clinical information are collected based on the BMO database registry. The specific analysis for this study was conducted based on the University of Texas MD Anderson institutional protocol PA18-0021, which was reviewed and approved by the University of Texas MD Anderson Cancer Center Institutional Review Board (MDACC IRB). Since the study was a retrospective tissue-based analysis, the need for informed consents were waved by MDACC IRB.

### FISH

FISH assay was performed on formalin-fixed, paraffin-embedded specimens using the Path-Vysion HER2 DNA Probe Kit (Abbott Laboratories). A dual probe (HER2 and CEP17) was used in all cases. The FISH result was interpreted as HER2-equivocal according to the 2013 ASCO/CAP guideline, and as group 4 according to the 2018 ASCO/CAP guideline, if the HER2/CEP17 ratio was <2 and the HER2 copy number was ≥4 and <6. If the HER2 FISH result was equivocal and the IHC result was 2+, another pathologist analyzed the same specimen by counting the signals in at least 20 cells that included the area with IHC 2+ staining. If the HER2 FISH result remained equivocal, the HER2 result was reclassified as negative.

## Immunohistochemistry

IHC staining was prospectively performed on freshly cut formalin-fixed, paraffin-embedded sections (5 μm) from the same specimen blocks that were used for the HER2 FISH study using the PATHWAY anti-HER2/neu (4B5) rabbit monoclonal antibody (Ventana Medical Systems, Inc.). If the IHC score was 0 or 1+, the HER2 status was reclassified as negative. If the IHC score was 3+, the HER2 status was reclassified as positive. If the IHC score was 2+, the HER2 FISH samples were sent for re-counting by a different pathologist by counting at least 20 cells that included the area with IHC 2+ staining.

## Clinical outcomes

The primary clinical outcomes of interest in this study were overall survival (OS) and event-free survival (EFS). For the primary cohort, survival was defined as the time from the biopsy until death from any cause for OS (except in 5 patients who had OS measured from surgery) and until recurrence or death, whichever occurred first, for EFS. For the recurrent/metastatic cohort, survival was defined as the time from the biopsy for recurrent disease or any biopsy confirmation of stage IV de novo cancer until death from any cause for OS and until progression (assessed clinically or pathologically) or death, whichever occurred first, for EFS. Patients were censored at the last follow-up if they were alive without any event (no recurrence, no progression, and no death).

## Statistical analysis

Data were summarized using standard descriptive statistics. Median, mean, and range where appropriate were used to describe continuous variables, and frequency and proportion were used to describe categorical variables. OS time and EFS time were estimated using the Kaplan-Meier method and compared between groups based on patient characteristics using the log-rank test. All computations were carried out in SAS 9.4 (SAS Institute Inc.).

# Results

## Patients

We identified 236 cases with HER2-equivocal FISH results per the 2013 ASCO/CAP guideline. Ninety-seven cases were excluded: 45 had a HER2-positive FISH result from a biopsy at an outside facility; 27 represented the second or greater HER2-equivocal FISH result in the same patient; 4 were HER2-equivocal FISH results from a non-breast cancer, all cancers of gastrointestinal origin; 3 occurred in patients with synchronous primary cancers; 14 were missing survival data; and 4 occurred in patients enrolled in a clinical trial of the E75 vaccine to boost anti-HER2 therapy. The remaining 139 cases were included, 90 in the primary cohort and 49 in the recurrent/metastatic cohort (Fig 1). In the primary cohort, 33 cases were treated with neoadjuvant therapy, and 57 were treated with adjuvant therapy. In the recurrent/metastatic cohort, the origin of the biopsied tissue was a locoregional recurrence in 3 cases, a distant recurrence in 32 cases, and stage IV de novo cancer in 14 cases.

Reasons for exclusion (97 cases):

- 45 HER2-positive FISH result from outside facility

- 27 multiple HER2-equivocal FISH results in same patient

- 4 HER2-equivocal FISH result from another cancer

- 3 synchronous primary cancer

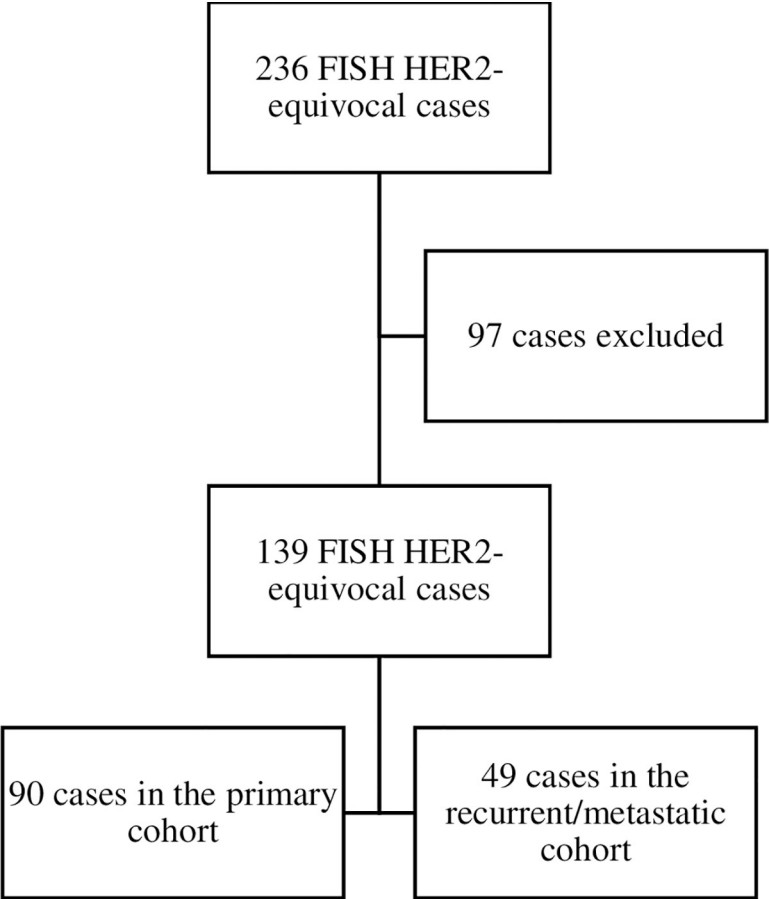

**Fig 1. FISH HER2-equivocal cases, excluded cases, and distribution into 2 cohorts.**

- 14 missing survival date

    4 enrolled in clinical trial of E75 vaccine
    FISH: fluorescence in situ hybridization. HER2: human epidermal growth factor receptor 2.
    Clinical and disease characteristics of the patients are summarized in Table 1. In the primary cohort, most patients were female, postmenopausal, or white, and many had an invasive ductal carcinoma, a grade 3 tumor, estrogen receptor–positive disease, or stage II disease. The median age was 52 years, the median FISH ratio was 1.3, the median HER2 copy number was 4.43. In the recurrent/metastatic cohort, patients had similar clinical characteristics as the primary cohort. Sixty five percent of samples in this cohort was biopsied from distant recurrence. The median age was 58 years, the median FISH ratio was 1.2, the median HER2 copy number was 4.43.

## Reclassification to HER2-negative status

After reclassification according to the 2018 ASCO/CAP guideline, 112 of the 139 cases (80.5%) were HER2-negative, 1 case (0.7%) was HER2-positive, and 26 cases (18.7%) had unknown HER2 status (Table 2). In the primary cohort, 94.4%, 1.1%, and 4.4% of cases were reclassified as HER2-negative, HER2-positive, and HER2-unknown, respectively, and in the recurrent/metastatic cohort, 55.1%, 0%, and 44.9% of cases were reclassified as HER2-negative, HER2-positive, and HER2-unknown, respectively (Table 2).

**Table 1. Clinical and disease characteristics.**

| Characteristic | Primary cohort (n = 90) | Recurrent/metastatic cohort (n = 49) |
|---|---|---|
| Age at diagnosis–median (range), y | 52 (32–91) | 58 (30–83) |
| Race/ethnicity–no. (%) | | |
| White | 58 (64) | 33 (67) |
| Asian | 4 (4) | 2 (4) |
| Black | 14 (16) | 7 (14) |
| Hispanic | 11 (12) | 4 (8) |
| Other | 3 (3) | 3 (6) |
| Sex–no. (%) | | |
| Female | 86 (96) | 49 (100) |
| Male | 4 (4) | |
| Menopausal status at diagnosis–no. (%) | | |
| Postmenopausal | 61 (71) | 28 (58) |
| Premenopausal | 25 (29) | 20 (42) |
| Unknown | 4 | 1 |
| Histology/cytology–no. (%) | | |
| Invasive ductal carcinoma | 82 (91) | 42 (86) |
| Invasive lobular carcinoma | 2 (2) | 2 (4) |
| Invasive mammary carcinoma | 3 (3) | 2 (4) |
| Metastatic carcinoma | 2 (2) | 3 (6) |
| Poorly differentiated carcinoma | 1 (1) | |
| Stage–no. (%) | | |
| I | 31 (35) | — |
| II | 35 (39) | — |
| III | 23 (26) | — |
| IV de novo | — | 14 (29) |
| Distant recurrence | — | 32 (65) |
| Locoregional recurrence | — | 3 (6) |
| Unknown | 1 | 0 |
| Tumor grade–no. (%) | | |
| 1 | 3 (3) | 0 |
| 2 | 36 (40) | 13 (81) |
| 3 | 50 (56) | 3 (19) |
| Unknown | 1 | 33 |
| Estrogen receptor status–no. (%) | | |
| Positive | 76 (84) | 41 (84) |
| Negative | 14 (16) | 8 (16) |
| Progesterone receptor status–no. (%) | | |
| Positive | 63 (71) | 22 (45) |
| Negative | 26 (29) | 27 (55) |
| Unknown | 1 | |
| FISH ratio–median (range) | 1.3 (1–2) | 1.2 (1–2) |
| HER2 copy number–median (range) | 4.4 (4–6) | 4.43 (4–6) |
| Ki-67 –median (range), % | 25 (2–95) | 22.5 (3–95) |

FISH: fluorescence in situ hybridization. HER2: human epidermal growth factor receptor 2.

**Table 2. HER2 status of the HER2-equivocal cases after reclassification according to 2018 ASCO/CAP guideline.**

| HER2 status after reclassification | No. (%) of cases | | |
|---|---|---|---|
| | Total | Primary cohort | Recurrent/metastatic cohort |
| Negative | 112 (81) | 85 (94) | 27 (55) |
| Positive | 1 (1) | 1 (1) | 0 (0) |
| Unknown | 26 (19) | 4 (4) | 22 (45) |

ASCO/CAP: American Society of Clinical Oncology/College of American Pathologists. HER2: human epidermal growth factor receptor 2.

In the cases in which the IHC score was 2+, prompting re-counting by an additional pathologist, the HER2 score remained equivocal. The IHC scores from the cases with HER2-equivocal results, overall and by cohort, are summarized in Table 3

## Treatment

In the primary cohort, 27.7% of cases were treated with adjuvant chemotherapy, 73.3% with adjuvant hormonotherapy, and 36.6% with neoadjuvant chemotherapy. Anthracycline-based chemotherapy was chosen in 72.0% of the cases in the adjuvant setting and in 93.9% of the cases in the neoadjuvant setting (Table 4). In the primary cohort, 5 (5.5%) cases were treated with anti-HER2 therapy, all of them from the group reclassified as HER2-negative according to the 2018 ASCO/CAP guideline (Fig 2). In the recurrent/metastatic cohort, 1 (2.0%) case was treated with anti-HER2 therapy; this case was also reclassified as HER2-negative according to the 2018 guideline (Fig 2). The chemotherapy backbone regimens used with the anti-HER2 therapy are described in S1 Table.

Of the 90 HER2-equivocal cases in the primary cohort, 85 (94.4%) cases were reclassified as HER2-negative according to the 2018 ASCO/CAP guideline. Five cases received anti-HER2 therapy, all in the HER2-negative group. After a median follow-up of 1.91 years, 5 progressions without death and 3 deaths had occurred in the primary cohort. HER2: human epidermal growth factor receptor 2. IHC: immunohistochemistry. w/: with. w/o: without.

## Clinical outcomes

In the primary cohort, after a median follow-up time of 1.91 years, 5 cases of recurrence without death and 3 deaths had occurred (Fig 2). In the recurrent/metastatic cohort, after a median follow-up time of 2.96 years, 20 cases of progression without death and 15 deaths had occurred (Fig 3).

Of the 49 HER2-equivocal cases in the recurrent/metastatic cohort, 27 (55.1%) cases were reclassified as HER2-negative according to the 2018 ASCO/CAP guideline. Only 1 case

**Table 3. HER2 IHC scores for the HER2-equivocal cases.**

| HER2 IHC score | No. (%) of cases | | |
|---|---|---|---|
| | Total | Primary cohort | Recurrent/metastatic cohort |
| 0 | 9 (6.4) | 6 (6.6) | 3 (6.1) |
| 1+ | 45 (32.3) | 34 (37.7) | 11 (22.4) |
| 2+ | 58 (41.7) | 45 (50.0) | 13 (26.5) |
| 3+ | 1 (0.7) | 1 (1.1) | 0 (0) |
| Unknown | 26 (18.7) | 4 (4.4) | 22 (44.9) |

HER2: human epidermal growth factor receptor 2. IHC: immunohistochemistry.

**Table 4. Treatment and residual cancer burden in the HER2-equivocal cases.**

| Characteristic | No. (%) of patients |
|---|:---:|
| *Primary cohort* | |
| Adjuvant hormonotherapy | |
| Yes | 66 (73.3) |
| No | 24 (26.6) |
| Exclusively | 26 (28.8) |
| Adjuvant chemotherapy | |
| Yes | 25 (27.7) |
| No | 65 (72.2) |
| Adjuvant chemotherapy agent | |
| Anthracycline-containing regimen | 18 (72.0) |
| Non–anthracycline-containing regimen | 7 (28.0) |
| Neoadjuvant chemotherapy | |
| Yes | 33 (36.6) |
| No | 57 (63.3) |
| Neoadjuvant chemotherapy agents | |
| Anthracycline-containing regimen | 31 (93.9) |
| Non–anthracycline-containing regimen | 2 (6.1) |
| Residual cancer burden class | |
| Pathologic complete response | 7 (25.0) |
| I | 1 (3.6) |
| I/II | 1 (3.6) |
| II | 12 (42.9) |
| III | 7 (25.0) |
| Unknown | 5 |
| Anti-HER2 therapy | |
| Yes | 5 (5.5) |
| No | 85 (94.4) |
| *Recurrent/metastatic cohort* | |
| Anti-HER2 therapy | |
| Yes | 1 (2.0) |
| No | 48 (98.0) |

HER2: human epidermal growth factor receptor 2.

received anti-HER2 therapy, in the HER2-negative group. After a median follow-up of 2.96 years, 20 progressions without death and 15 deaths had occurred in the recurrent/metastatic cohort. HER2: human epidermal growth factor receptor 2. IHC: immunohistochemistry. w/: with. w/o: without.

In the primary cohort, the median EFS time was not reached, and the EFS rate at 2 years was 90% in patients who did not receive anti-HER2 therapy and 100% in those who received anti-HER2 therapy. In the recurrent/metastatic cohort, the median EFS time was 1.41 years, and the EFS rate at 1 year was 51% in patients who did not receive anti-HER2 therapy and 100% in those who received anti-HER2 therapy (Fig 4). There was no association between the use of anti-HER2 therapy and EFS in either the primary cohort (p = 0.49) or the recurrent/ metastatic cohort (p = 0.58). In the primary cohort, the median OS time was not reached, and the OS rate at 2 years was 94% in patients who did not receive anti-HER2 therapy and 100% in those who received anti-HER2 therapy. In the recurrent/metastatic cohort, the median OS

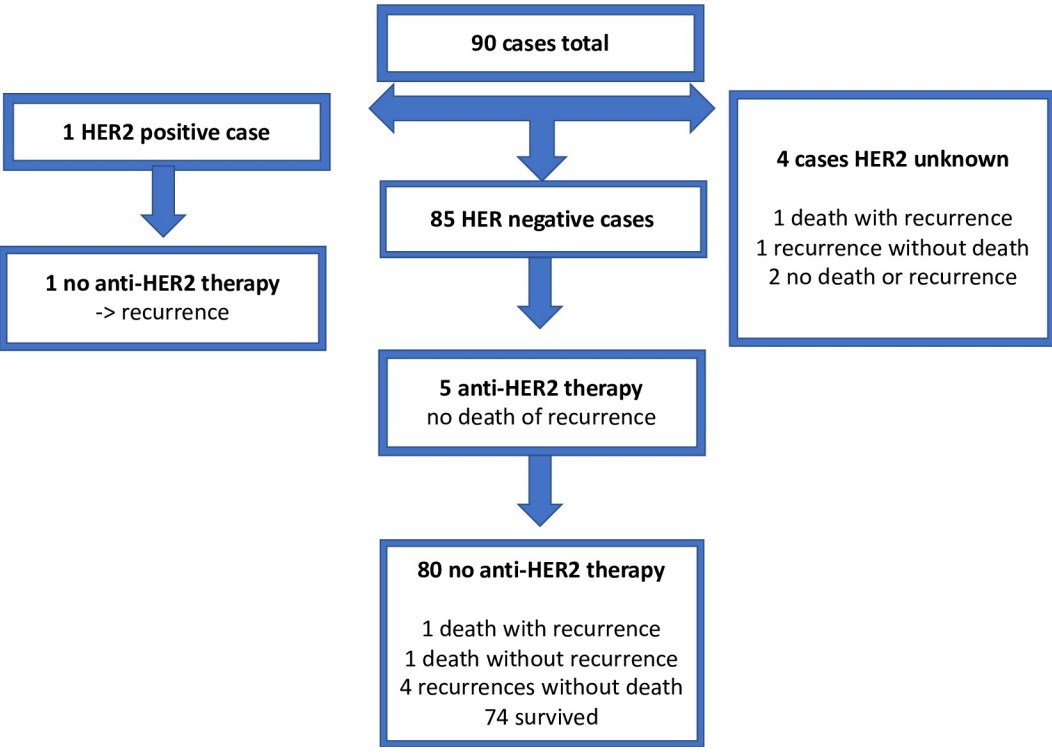

**Fig 2. Anti-HER2 therapy and events in the primary cohort.**

time was 6.3 years, and the OS rate at 1 year was 81% in patients who did not receive anti-HER2 therapy and 100% in those who received anti-HER2 therapy (Fig 5). There was no association between the use of anti-HER2 therapy and OS in either the primary cohort (p = 0.67) or the recurrent/metastatic cohort (p = 0.67).

## Discussion

In this study, most of the HER2-equivocal breast cancer cases were reclassified as HER2-negative per the 2018 HER2 ASCO/CAP guideline. No association was found between anti-HER2 therapy and OS or EFS among cases with HER2-negative status according to the 2018 guideline.

Interestingly, the proportion of HER2-equivocal cases reclassified as HER2-negative was lower than previously reported. Xu et al [16] and Liu et al [17] previously reported that 100% (44 of 44) and 98.8% (173 of 175) of HER2-equivocal cases, respectively, were reclassified as HER2-negative according to the 2018 ASCO/CAP guideline. In our study, only 80.5% of HER2-equivocal cases were reclassified as HER2-negative. This discrepancy may be due to the population we chose for the study. Xu et al included patients in whom the biopsy specimens originated exclusively from primary tumors, while Liu et al included patients irrespective of the origin of the biopsy specimen. These populations are both different from our patient population, which included a high proportion of patients in whom the biopsy specimen originated from the site of recurrent/metastatic disease. In our primary cohort, the proportion of HER2-equivocal cases reclassified as HER2-negative was similar (94.4%) to the proportions in both previous studies. However, in our recurrent/metastatic cohort, we faced the challenge of insufficient cytological material for further analysis, which resulted in a higher rate of

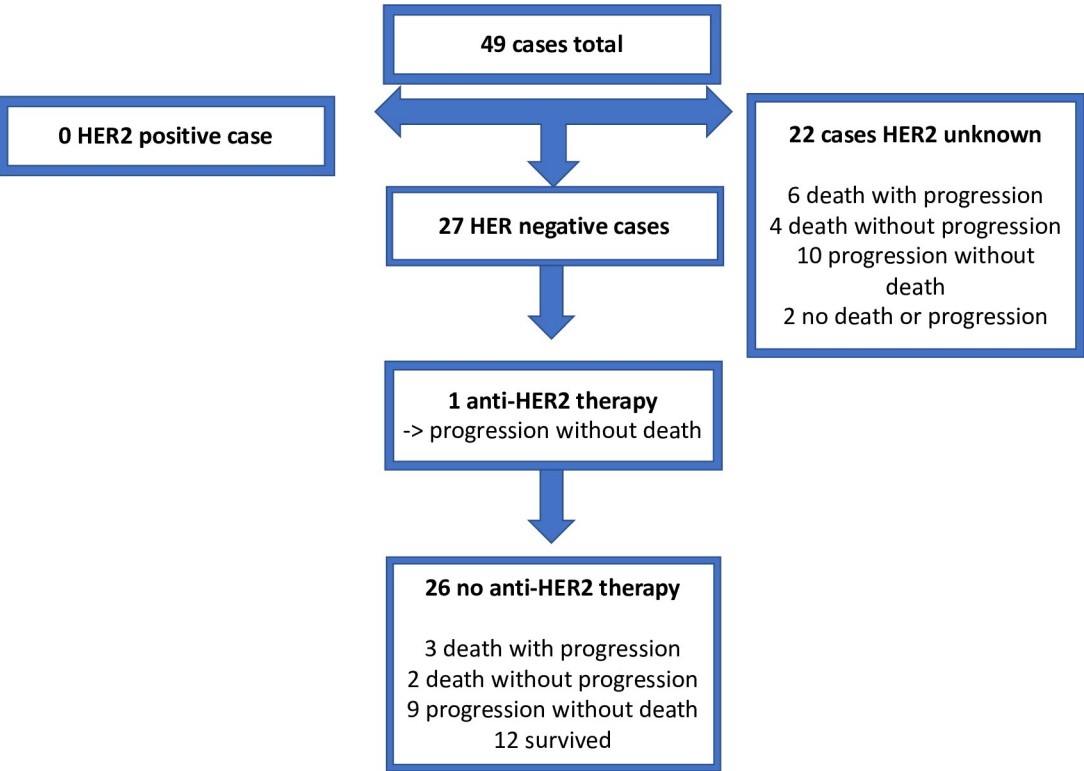

**Fig 3. Anti-HER2 therapy and events in the recurrent/metastatic cohort.**

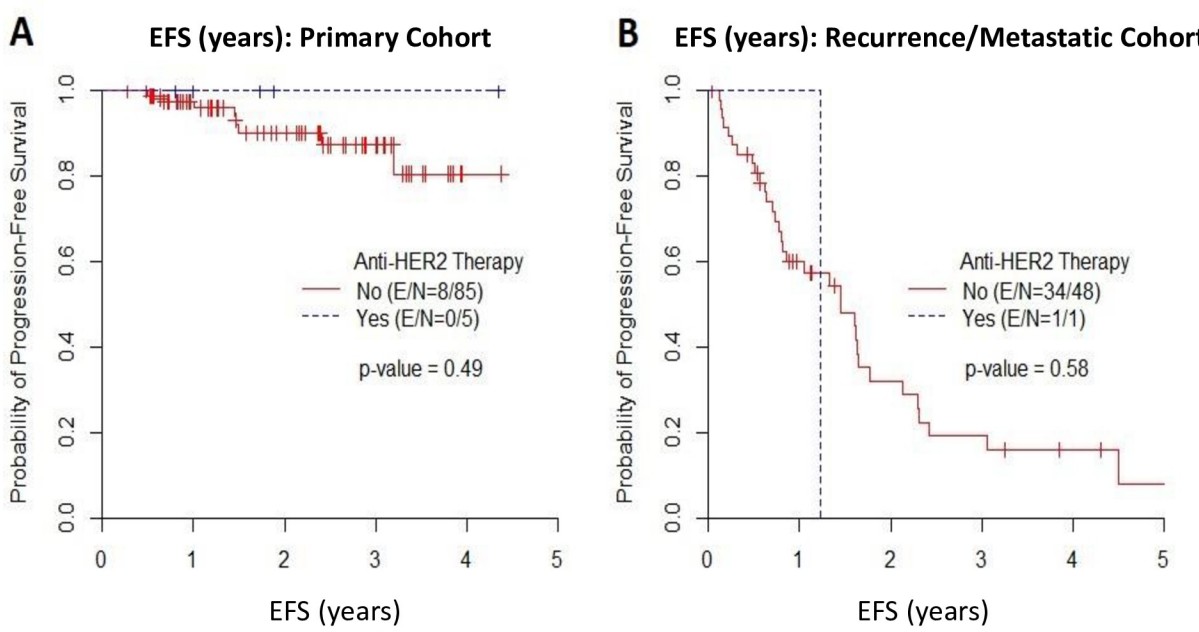

**Fig 4. Event-Free Survival (EFS).** Kaplan-Meier estimates of EFS in (A) the primary cohort and (B) the recurrent/metastatic cohort.

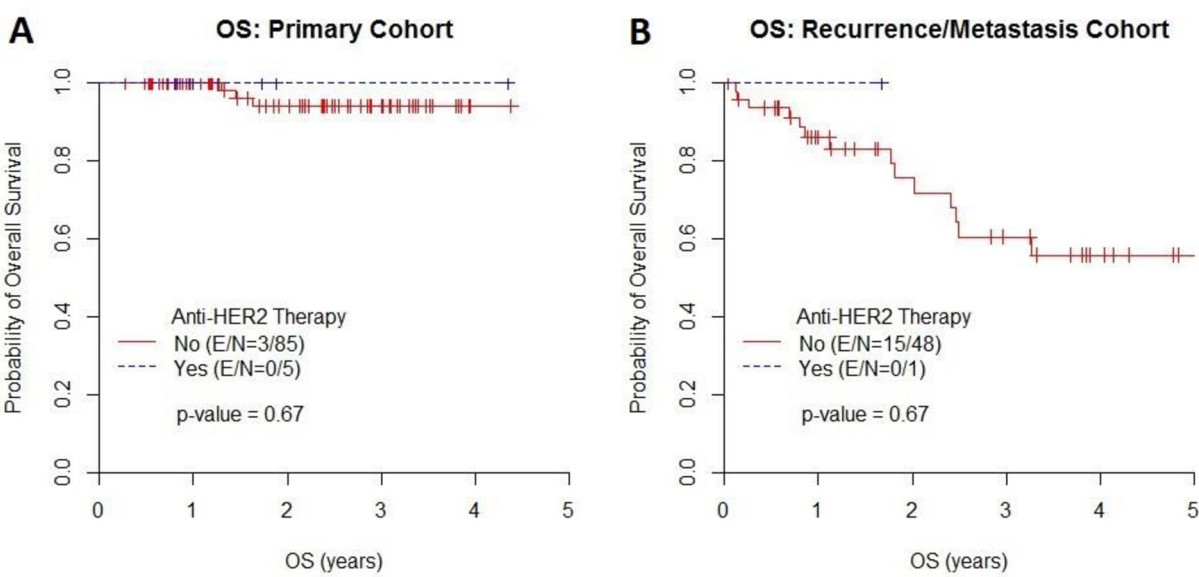

**Fig 5. Overall Survival (OS).** Kaplan-Meier estimates of OS in (A) the primary cohort and (B) the recurrent/metastatic cohort.

HER2-unknown cases on reclassification (44.9% in the recurrent/metastatic cohort vs. 4.4% in the primary cohort). This difference of HER2-unkonw cases between two guidelines might also be due to aggressive re-counting HER2 signals by our dedicated breast pathologists. Indeed, the inter-institutional HER2 FISH discrepancy has been previously reported [18].

To the best of our knowledge, this study is the first to analyze the relationship between EFS and OS and anti-HER2 therapy use in HER2-equivocal breast cancer cases. Unfortunately, the analysis did not show a significant difference in either OS or EFS according to receipt of anti-HER2 therapy. However, the statistical analysis was limited by the small number of cases (6 total) treated with anti-HER2 therapy and the rare cases of recurrence and progression. The reason most of our physicians opted not to use anti-HER2 therapy for this cohort of patients is not clear; however, we speculate that physicians were influenced by studies showing that HER2-equivocal cases may behave like HER2-negative cases [19, 20] and therefore ignored the 2013 ASCO/CAP recommendations to treat HER2-equivocal cases with anti-HER2 therapy [2]. In the primary cohort, the prognosis of the HER2-equivocal cases was similar to that of historical HER2-negative cases [21]. Indeed, the EFS rate at 2 years of ≥90% would have been impossible without anti-HER2 therapy if our cases had exhibited the typical biology of HER2-positive breast cancers [21]. Tong et al. recently reported that the clinical characteristics of HER2-equivocal cases were more similar to those of HER2-negative cases than to those of HER2-positive cases, which also supports this hypothesis [22].

We excluded 97 of 236 cases with HER2-equivocal FISH results from our dataset. In 45 cases, the exclusion was due to an outside HER2-positive result by either IHC or by FISH. All 45 of these cases were treated with anti-HER2 therapy, and these cases could have affected the outcome of our study. Griggs et al. compared the discordance in HER2 results between original and central laboratories and found a HER2 discordance rate of 26% by IHC and of 6% by FISH (performed only if HER2 had an IHC score of 2+) [23]. Moreover, in a recent study [17], chromosome 17 polysomy in HER2-equivocal cases [24] was responsible for a large proportion of changes in HER2 status between classification according to the 2013 and 2018 ASCO/CAP guidelines. Another potential explanation for the discrepancy in results between our

laboratory and outside laboratories in these 45 cases is tumor heterogeneity [25, 26]. The significance of these discrepancies needs to be studied further because some patients could have received unnecessary anti-HER2 therapy and thereby been exposed to the associated risk of cardiac adverse events.

Our study has several limitations, including the highly selected group of patients included, the small number of cases, the short follow-up time, and the retrospective nature of the study. The inclusion of the patients based on HER2 FISH results could have resulted in inclusion of a high proportion of patients with a HER2 IHC score of 2+ because a HER2 IHC score of 2+ is the typical situation in which physicians proceed with a FISH test. However, approximately 40% of the patients included had a HER2 IHC score other than 2+. Another limitation is that the study was done in only 1 center, which could have been a reason for a low number of patients receiving anti-HER2 therapy. As note, one goal of our study was to address the difference between 2013 ASCO/CAP guideline vs 2018 ASCO/CAP guideline about the HER2 status and see how this change could impact the therapeutic efficacy. The fact that most of them are re-classified as HER2 negative, and even small population did not show the HER2 agent therapy benefit supports that new 2018 ASCO/CAP guideline may be more effective in defining the clinically HER2 positive vs negative groups.

Despite these limitations, our results corroborate the changes in HER2 classification in the 2018 ASCO/CAP guideline and do not support the use of anti-HER2 therapy in the previously classified HER2-equivocal group based on 2013 criteria. A prospective, multicenter randomized clinical trial with a larger cohort of patients, and a group (group 1–4 in 2018 criteria) based re-analysis is warranted to confirm our observation.

## Supporting information

**S1 Table. Backbone chemotherapy regimens containing anti-HER2 therapy.** (DOCX)

## Acknowledgments

We thank Stephanie Deming of Editing Services, Research Medical Library, MD Anderson Cancer Center, for editing the manuscript.

## Author Contributions

**Conceptualization:** Aysegul A. Sahin, Bora Lim.

**Data curation:** James Crespo, Hongxia Sun, Jimin Wu, Qing-Qing Ding, Guilin Tang, Melissa K. Robinson, Hui Chen, Aysegul A. Sahin, Bora Lim.

**Formal analysis:** James Crespo, Hongxia Sun, Jimin Wu, Guilin Tang, Melissa K. Robinson, Aysegul A. Sahin, Bora Lim.

**Funding acquisition:** Hongxia Sun, Aysegul A. Sahin, Bora Lim.

**Investigation:** James Crespo, Jimin Wu, Qing-Qing Ding, Guilin Tang, Melissa K. Robinson, Hui Chen, Bora Lim.

**Methodology:** James Crespo, Hongxia Sun, Jimin Wu, Qing-Qing Ding, Guilin Tang, Hui Chen, Aysegul A. Sahin, Bora Lim.

**Project administration:** Aysegul A. Sahin, Bora Lim.

**Resources:** Aysegul A. Sahin, Bora Lim.

**Software:** James Crespo, Hongxia Sun, Jimin Wu, Melissa K. Robinson.

**Supervision:** Aysegul A. Sahin, Bora Lim.

**Validation:** James Crespo, Jimin Wu, Qing-Qing Ding, Guilin Tang, Hui Chen, Bora Lim.

**Writing – original draft:** James Crespo, Bora Lim.

**Writing – review & editing:** Aysegul A. Sahin, Bora Lim.

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
