## [Decision Letter · Decision Letter 0]

15 Sep 2020

PONE-D-20-23814

Rate of reclassification of HER2-equivocal breast cancer cases to HER2-negative per the 2018 ASCO/CAP guidelines and response of HER2-equivocal cases to anti-HER2 therapy

PLOS ONE

Dear Dr. Lim,

Thank you for submitting your manuscript to PLOS ONE. After careful consideration, we feel that it has merit but does not fully meet PLOS ONE’s publication criteria as it currently stands. Therefore, we invite you to submit a revised version of the manuscript that addresses the points raised during the review process.

We look forward to receiving your revised manuscript.

Kind regards,

Paul J van Diest

Academic Editor

PLOS ONE

Journal Requirements:

2.Thank you for including your ethics statement: 'The specific analysis for this study was conducted based on MD Anderson institutional protocol PA18-0021, which was reviewed and approved by the IRB. Since the study was a retrospective tissue-based analysis, no informed consent was obtained specifically for this study'.   

(a) Please amend your current ethics statement to include the full name of the ethics committee/institutional review board(s) that approved your specific study.  

(b) Once you have amended this/these statement(s) in the Methods section of the manuscript, please add the same text to the “Ethics Statement” field of the submission form (via “Edit Submission”).

3. In ethics statement in the manuscript and in the online submission form, please provide additional information about the patient records/samples used in your retrospective study. Specifically, please ensure that you have discussed whether all data/samples were fully anonymized before you accessed them and/or whether the IRB or ethics committee waived the requirement for informed consent. If patients provided informed written consent to have data/samples from their medical records used in research, please include this information.

4. To comply with PLOS ONE submission guidelines, in your Methods section, please provide additional information regarding your statistical analyses. For more information on PLOS ONE's expectations for statistical reporting, please see https://journals.plos.org/plosone/s/submission-guidelines.#loc-statistical-reporting.

5.Thank you for stating the following financial disclosure:

 [The funders had no role in study design, data collection and analysis, decision to publish, or preparation of the manuscript.].

Reviewers' comments:

Reviewer's Responses to Questions

**Comments to the Author**

1. Is the manuscript technically sound, and do the data support the conclusions?

Reviewer #1: Yes

Reviewer #2: Partly

2. Has the statistical analysis been performed appropriately and rigorously? 

Reviewer #1: Yes

Reviewer #2: Yes

3. Have the authors made all data underlying the findings in their manuscript fully available?

Reviewer #1: Yes

Reviewer #2: Yes

4. Is the manuscript presented in an intelligible fashion and written in standard English?

Reviewer #1: Yes

Reviewer #2: Yes

5. Review Comments to the Author

Reviewer #1: This is a well written article on category/group 4 of the HER2 dual ISH assessment results as defined by the 2018 ASCO/CAP guidelines. Most of the cases classified as equivocal on the basis of the 2013 recommendations turned out to be negative per definitions of the 2018 recommendations. The limitations of the study are listed, and two of these are the virtual lack of targeted treatment in this subpopulation, and the low frequency of events overall.

The references are not uniformly presented (and therefore cannot follow a uniform instruction set for authors); e.g. both capitals and lower case letters for unabbreviated journal names; full confusion with the author names in item 18...

Reviewer #2: This is a retrospective single center study investigating the rate of reclassification of so-called HER2-equivocal breast cancers (according to ASCO/CAP guidelines of 2013) to either HER2-negative or HER2-positive breast cancers according to revised ASCO/GAP guidelines of 2018. As the benefit of anti-HER2 therapy is obvious in HER2-positive disease but remains unclear in HER2-equivocal breast cancer, it is important to learn if anti-HER2 therapy is valuable in cases who are to be reclassified as HER2-positive according to new guidelines.

The study has been conducted in a well-described manner and FISH analysis and IHC staining are according to international standards.

The authors found that only 80.5% of cases were reclassified as HER2-negative. However, in the results it remains unclear why a considerable proportion of cases (n=26 19%) could not be reclassified to either HER2-positive or negative breast cancer. The definition of unknown HER2 status is lacking: was it technically impossible to reclassify or were insufficient tumour cells available for testing? More importantly, only 1 out of 139 cases was reclassified as HER2-positive.

The impact of anti-HER2 treatment in patients with equivocal HER2-testing in this series remains unclear, as only 1 patient was reclassified as HER2-positive and 19% was reclassified as HER2-unknown, while none of these patients received anti-HER2 treatment. From the NSABP-42 study it is already known that anti-HER2 therapy of HER21+ or 2+ breast cancer as no additional benefit.

This study does therefore not allow to make any conclusions on the impact of anti-HER2 therapy in HER2-equivocal breast cancers, as the number of patients treated with anti-HER2, therapy is too low (n=1 in early breast cancer and n=5 in metastatic breast cancer). Besides, the median follow up (<2 years) is too short to detect any potential benefit both in early breast cancer as well as in metastatic breast cancer as recurrences occur up to 5-10 years after primary diagnosis and median survival in metastatic breast cancer responding to anti-HER2 therapy exceeds 4 years (Cleopatra study).

Reference 7 on a phase 1 study with trastuzumab deruxtecan could be replaced by a phase 2 study by Modi S et al, NEJM 2020;382:610-21.

In conclusion, the study is well written and well performed, but the conclusion on the impact of anti-HER2 therapy in HER2-equivocal breast cancers can not be made.

6. PLOS authors have the option to publish the peer review history of their article (what does this mean?). If published, this will include your full peer review and any attached files.

Reviewer #1: **Yes: **Gábor CSERNI

Reviewer #2: **Yes: **Carolina Smorenburg

---

## [Author Response · Author response to Decision Letter 0]

6 Oct 2020

Sep 15, 2020

Dear Dr. Paul J van Diest

Academic Editor

PLOS ONE

Re: PONE-D-20-23814

Rate of reclassification of HER2-equivocal breast cancer cases to HER2-negative per the 2018 ASCO/CAP guidelines and response of HER2-equivocal cases to anti-HER2 therapy

We appreciate the kind and thorough review from Drs Cserni, and Smorenburg. 

Here in this revised version, we tried to address all the concerns from the reviewers. The line by line responses to each concern from the reviewers are listed below. We sincerely hope we have either clarified or fixed the areas of concerns. 

Reviewer #1: This is a well written article on category/group 4 of the HER2 dual ISH assessment results as defined by the 2018 ASCO/CAP guidelines. Most of the cases classified as equivocal on the basis of the 2013 recommendations turned out to be negative per definitions of the 2018 recommendations. The limitations of the study are listed, and two of these are the virtual lack of targeted treatment in this subpopulation, and the low frequency of events overall.

We agree that this is a concern and a limitation. However, we believe that negative findings in such case could still be informative to the field, and therefore our study would further encourage the wider studies to continue to improve our targeted therapeutic strategy towards HER2. We tried to elaborate this limitation further in the discussion section.

The references are not uniformly presented (and therefore cannot follow a uniform instruction set for authors); e.g. both capitals and lower case letters for unabbreviated journal names; full confusion with the author names in item 18...

Thank you for pointing this out. We realized that the endnote program did not fix all the references into correct format and we overlooked these - that created errors in several references. We made sure to revise all the references to ensure all the references follow the same format in this revised version. 

Reviewer #2: This is a retrospective single center study investigating the rate of reclassification of so-called HER2-equivocal breast cancers (according to ASCO/CAP guidelines of 2013) to either HER2-negative or HER2-positive breast cancers according to revised ASCO/GAP guidelines of 2018. As the benefit of anti-HER2 therapy is obvious in HER2-positive disease but remains unclear in HER2-equivocal breast cancer, it is important to learn if anti-HER2 therapy is valuable in cases who are to be reclassified as HER2-positive according to new guidelines.

The study has been conducted in a well-described manner and FISH analysis and IHC staining are according to international standards.The authors found that only 80.5% of cases were reclassified as HER2-negative. However, in the results it remains unclear why a considerable proportion of cases (n=26 19%) could not be reclassified to either HER2-positive or negative breast cancer. The definition of unknown HER2 status is lacking: was it technically impossible to reclassify or were insufficient tumour cells available for testing? More importantly, only 1 out of 139 cases was reclassified as HER2-positive.

In part, our MD Anderson pathology group follows rigorous exercise when the HER2 is reclassified. Many of our reclassified patients in the recurrent/metastatic group, we obtained additional cytology material to confirm the HER2 FISH status (as mentioned in the page 19, section “ In our primary cohort, the proportion of HER2-equivocal cases reclassified as HER2-negative was similar (94.4%) to the proportions in both previous studies. However, in our recurrent/metastatic cohort, we faced the challenge of insufficient cytological material for further analysis, which resulted in a higher rate of HER2-unknown cases on reclassification (44.9% in the recurrent/metastatic cohort vs. 4.4% in the primary cohort). This difference of HER2-unkonw cases between two guidelines might also be due to aggressive re-counting HER2 signals by our dedicated breast pathologists. Indeed, the inter-institutional HER2 FISH discrepancy has been previously reported (18).” Of course, we will not be able to examine the same in the other studies to carefully evaluate the discrepancy among studies, yet we believe this may be the biggest cause of the difference. We hope our explanation that was already included in the previous manuscript is sufficient to rationalize this. 

The impact of anti-HER2 treatment in patients with equivocal HER2-testing in this series remains unclear, as only 1 patient was reclassified as HER2-positive and 19% was reclassified as HER2-unknown, while none of these patients received anti-HER2 treatment. From the NSABP-42 study it is already known that anti-HER2 therapy of HER21+ or 2+ breast cancer as no additional benefit.

We agree. This is one of the major limitations of our study. We believe this was mainly due to reclassification to HER2 negative group at the end of our analysis to re-classify them. As note, one goal of our study was to address the difference between 2013 ASCO/CAP guideline vs 2018 ASCO/CAP guideline about the HER2 status and see how this change could impact the therapeutic efficacy. The fact that most of them are re-classified as HER2 negative, and even small population did not show the HER2 agent therapy benefit supports that new 2018 ASCO/CAP guideline may be more effective in defining the clinically HER2 positive vs negative groups. We added this section to the discussion section as well. 

This study does therefore not allow to make any conclusions on the impact of anti-HER2 therapy in HER2-equivocal breast cancers, as the number of patients treated with anti-HER2, therapy is too low (n=1 in early breast cancer and n=5 in metastatic breast cancer). Besides, the median follow up (<2 years) is too short to detect any potential benefit both in early breast cancer as well as in metastatic breast cancer as recurrences occur up to 5-10 years after primary diagnosis and median survival in metastatic breast cancer responding to anti-HER2 therapy exceeds 4 years (Cleopatra study).

In conclusion, the study is well written and well performed, but the conclusion on the impact of anti-HER2 therapy in HER2-equivocal breast cancers can not be made.

I think maybe our statement here gave incorrect message about the anti-HER2 therapy in equivocal cancer – the equivocal group here we mention is based on 2013. To reduce the confusion, we have revised the conclusion sentence that is currently written as “Despite these limitations, our results corroborate the changes in HER2 classification in the 2018 ASCO/CAP guideline and do not support the use of anti-HER2 therapy in the HER2-equivocal group.” -> to “Despite these limitations, our results corroborate the changes in HER2 classification in the 2018 ASCO/CAP guideline and do not support the use of anti-HER2 therapy in the previously classified HER2-equivocal group based on 2013 criteria.”

Reference 7 on a phase 1 study with trastuzumab deruxtecan could be replaced by a phase 2 study by Modi S et al, NEJM 2020;382:610-21.

Thank you. We have replaced the reference and revised all of our reference to ensure that we follow a correct formatting.

Sincerely,

Bora Lim, M.D.

---

## [Editor Report · Decision Letter 1]

21 Oct 2020

Rate of reclassification of HER2-equivocal breast cancer cases to HER2-negative per the 2018 ASCO/CAP guidelines and response of HER2-equivocal cases to anti-HER2 therapy

PONE-D-20-23814R1

Dear Dr. Lim,

We’re pleased to inform you that your revised manuscript has been judged scientifically suitable for publication and will be formally accepted for publication once it meets all outstanding technical requirements.

Kind regards,

Paul J van Diest

Academic Editor

PLOS ONE

---

## [Editor Report · Acceptance letter]

4 Nov 2020

PONE-D-20-23814R1 

Rate of reclassification of HER2-equivocal breast cancer cases to HER2-negative per the 2018 ASCO/CAP guidelines and response of HER2-equivocal cases to anti-HER2 therapy 

Dear Dr. Lim:

I'm pleased to inform you that your manuscript has been deemed suitable for publication in PLOS ONE. Congratulations! Your manuscript is now with our production department. 

Kind regards, 

on behalf of

Dr. Paul J van Diest 

Academic Editor

PLOS ONE